# Ins-DetCLIP: Aligning Detection Model to Follow Human-Language Instruction

**Renjie Pi**[1,*], **Lewei Yao**[1,*], **Jianhua Han**[2], **Xiaodan Liang**[3], **Wei Zhang**[2], **Hang Xu**[2]
[1]Hong Kong University of Science and Technology   [2]Huawei Noah's Ark Lab
[3]Sun Yat-sen Unversity and MBZUAI

## Abstract

This paper introduces Instruction-oriented Object Detection (IOD), a new task that enhances human-computer interaction by enabling object detectors to understand user instructions and locate relevant objects. Unlike traditional open-vocabulary object detection tasks that rely on users providing a list of required category names, IOD requires models to comprehend natural-language instructions, contextual reasoning, and output the name and location of the desired categories. This poses fresh challenges for modern object detection systems. To develop an IOD system, we create a dataset called IOD-Bench, which consists of instruction-guided detections, along with specialized evaluation metrics. We leverage large-scale language models (LLMs) to generate a diverse set of instructions (8k+) based on existing public object detection datasets, covering a wide range of real-world scenarios. As an initial approach to the IOD task, we propose a model called Ins-DetCLIP. It harnesses the extensive knowledge within LLMs to empower the detector with instruction-following capabilities. Specifically, our Ins-DetCLIP employs a visual encoder (i.e., DetCLIP, an open-vocabulary detector) to extract object-level features. These features are then aligned with the input instructions using a cross-modal fusion module integrated into a pre-trained LLM. Experimental results conducted on IOD-Bench demonstrate that our model consistently outperforms baseline methods that directly combine LLMs with detection models. This research aims to pave the way for a more adaptable and versatile interaction paradigm in modern object detection systems, making a significant contribution to the field.

## 1 Introduction

Object detection serves as one of the most important tasks in the field of computer vision, with its influence spanning across numerous applications, including robotics (Bai et al., 2020; Karaoguz & Jensfelt, 2019; Zhang et al., 2019; Shridhar et al., 2022), autonomous driving (Liu et al., 2021a; Feng et al., 2021; Qian et al., 2022), and surveillance (Zhou et al., 2019; Ullah et al., 2021; Santhosh et al., 2020). The recent advent of powerful multi-modal models, such as CLIP (Radford et al., 2021), has ushered in a new era of open-vocabulary detection (OVD) algorithms (Gu et al., 2021b; Zhong et al., 2022; Li et al., 2021a; Yao et al., 2022a; Zhou et al., 2022). These models overcome the limitations of predefined class labels and enable recognition of a wide range of user-provided object classes. Despite the advancements, contemporary open-vocabulary object detectors still have limitations in terms of human-computer interaction. Specifically, they rely on users providing a comprehensive list of object classes, which may not be practical in real-world scenarios. In such cases, it is preferable for models to directly understand user instructions to identify and locate relevant objects. For example, a domestic robot should be able to recognize targets like "beverages," "drinks," and "cups" when instructed to "hand me a cup of drink". This observation motivates our pursuit of a more flexible human-computer interaction paradigm tailored to modern object detection systems.

The advent of large-scale language models (LLMs) (Brown et al., 2020; OpenAI., 2023; Touvron et al., 2023; Vicuna., 2023) has revolutionized user interactions by enabling natural language-based communication with models. Inspired by this breakthrough, we aim to establish natural language interaction with object detectors. This paper introduces Instruction-oriented Object Detection (IOD),

---

*Equal Contribution.

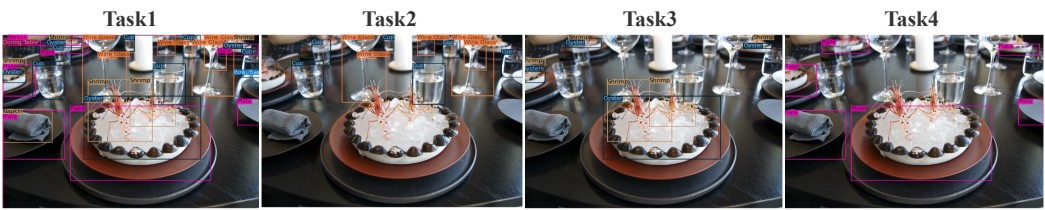

| Task1 | Task2 | Task3 | Task4 |
| :---: | :---: | :---: | :---: |
| **Identify all visible objects.** | **Detect cup and wine glass.** | **Discover all the seafood.** | **Clean dishes after dinner.** |

Figure 1: Visualization of the constructed 4 types of instructions. The first task aims to identify all visible objects in the image; the second task queries specific object categories; the third task retrieves the objects that are under a mutual parent class; the fourth task is to identify objects that are related to a daily life requirement.

a novel task that requires the model to understand user requirements and accurately localize relevant objects. IOD can be seen as an advanced extension of open-vocabulary object detection, where the detector must comprehend user-provided natural language instructions and infer the corresponding objects, surpassing the conventional approach which detect objects based solely on user-supplied class names. The IOD task entails more sophisticated reasoning and comprehension capabilities, introducing fresh challenges to the field of object detection.

To facilitate the proposed Instruction-oriented Object Detection (IOD) task, we address the lack of existing public datasets designed for this challenge. We create an instruction-guided detection dataset termed *IOD-Bench* and develop corresponding evaluation metrics. Our approach involves leveraging the class names provided by the Objects365 dataset (Shao et al., 2019) and utilizing a powerful LLM (OpenAI., 2022) to generate instructions related to these classes. To ensure a diverse and realistic set of instructions that cover various everyday scenarios, we design instructions with multiple purposes. These instructions can express specific goals that require the model to reason and detect relevant objects, or directly request the detection of specific categories. Through querying the LLM and applying post-processing filters, we curate approximately 8,000 instructions, allocating a portion for training and the remaining for evaluation. To ensure accurate evaluations in cases where the model produces correct outputs for instruction-related classes, but those outputs do not precisely match the class names present in the Objects365 dataset, we modify the native evaluation procedure and address the potential issues that may arise.

Then, we introduce a novel model called Ins-DetCLIP that integrates state-of-the-art open-vocabulary detectors (OVD) with large-scale language models, specifically tailored for the Instruction Object Detection (IOD) task. The training is conducted in a two-phase manner. In the initial phase, we pre-train the OVD model following the training pipeline of DetCLIP(Yao et al., 2022b). This pre-training enables the model to learn a visual encoder capable of extracting fine-grained visual features and accurately localizing objects, laying the foundation for robust object detection. In the subsequent phase, we seamlessly integrate a large-scale language model (LLM), which empowers the model with the ability to comprehend human instructions and perform instruction-based object detection. The model is trained with our curated dataset to generate outputs for each object conditioned on the provided instruction. We design a simple yet effective mechanism where the model predicts the corresponding category name if the instruction matches the object, while predicting the word "other" otherwise. During inference, objects labeled as "other" are filtered out, and only the objects of interest, as guided by the instruction, are selected. To establish the effectiveness of our proposed method, we further design several baseline methods for comparison. For instance, we leverage the open-source large-scale vision-language models, *i.e.*, BLIP-2 (Li et al., 2023a), MiniGPT4 (Zhu et al., 2023) or LLAVA (Liu et al., 2023a), to identify object categories related to the instructions in images. Subsequently, we utilize OVD models such as DetCLIP (Yao et al., 2022b) to detect objects belonging to these identified categories.

To evaluate the performance of our proposed approach, we conduct comprehensive experiments by comparing it with the aforementioned baselines on our constructed dataset. Our Ins-DetCLIP demonstrates significant superiority over the baselines, e.g., obtaining the 10.0%/9.7% average AP improvement on in-domain/out-domain instructions, showcasing its effectiveness and potential for IOD tasks. Furthermore, due to the exceptional generative capability, Ins-DetCLIP is also able to achieve the state-of-the-art performance on dense captioning benchmark(Johnson et al., 2016). This

demonstrates the model's prowess in both dense prediction and generation tasks, further showcasing its versatility and effectiveness. We wish this work can pave the way for a more adaptable and versatile human-computer interaction paradigm in modern object detection systems.

## 2 RELATED WORK

**Open-Vocabulary Object Detection** has recently provided a pragmatic framework for recognizing objects of unlimited categories. Thanks to the success of vision-language pre-training, contemporary research (Zang et al., 2022; Xie & Zheng, 2021; Gu et al., 2021a; Zhong et al., 2022) has put forth the idea of leveraging the wisdom from a pre-trained VL model (e.g., CLIP (Radford et al., 2021)) within a detection context. Another promising approach utilizes a broader dataset for training. For example, (Gao et al., 2021; Li et al., 2021b; Fontanel et al., 2022; Inkawhich et al., 2022; Yao et al., 2022b) have leveraged easy-to-access image-text pairs to expand the training domain's coverage through a pseudo labeling mechanism. Detclipv2(Yao et al., 2023) proposes a maximum word-region similarity between region proposals and textual words to guide the contrastive objective. XDETR (Cai et al., 2022) combines the conventional contrastive learning in VLP (Radford et al., 2021; Jia et al., 2021) to perform image-to-text alignment. Detic (Zhou et al., 2022), on the other hand, sets out to tackle a large-vocabulary detection challenge by assigning classification labels directly to the maximum-sized region proposals. Based on DINO (Zhang et al., 2022a; Liu et al., 2023b), OpenSeeD (Zhang et al., 2023) try to jointly learn from different segmentation and detection datasets. However, none of these methods consider detection based on human-language instructions, e.g., "find me some drinks on the desk", which is important for the future applications, like in robotics (Bai et al., 2020; Karaoguz & Jensfelt, 2019; Zhang et al., 2019; Shridhar et al., 2022) and autonomous driving (Liu et al., 2021a; Feng et al., 2021; Qian et al., 2022).

**Multi-Modality Large Language Model.** With the rise of text-based large language models (LLMs (Zeng et al., 2022; Zhang et al., 2022b; Chowdhery et al., 2022)), such as Flan (Chung et al., 2022), ChatGPT (OpenAI., 2022), and LLaMA (Touvron et al., 2023), many strategies have emerged that fine-tune these models to align visual language by leveraging visual-text data. For example, Flamingo (Alayrac et al., 2022) introduces a computation-efficient approach to vision-language pre-training, making use of frozen pre-trained image encoders and LLMs with gated cross-attention. BLIP-2 (Li et al., 2023b) propose the Q-former to efficiently align the visual and the text features with Flan-T5 (Chung et al., 2022) and OPT (Zhang et al., 2022b). Building on the groundwork of BLIP-2 (Li et al., 2023a), MiniGPT-4 (Zhu et al., 2023) employs a more powerful LLM, named Vicuna (Vicuna., 2023), to achieve advanced multi-modal generation capabilities. InstructBLIP (Dai et al., 2023) further enables the image encoder to extract different visual features given different instructions. In addition, LLaVA (Liu et al., 2023a) represents the pioneering effort to utilize the language-only GPT-4 (OpenAI., 2023) model to generate multi-modal data. However, none of these work considers predicting precise location (i.e., bounding-box coordinates) for each concepts explicitly or implicitly mentioned in the instruction.

## 3 IOD-BENCH: AN INSTRUCTION-ORIENTED OBJECT DETECTION DATASET

To facilitate an Instruction-oriented Object Detection (IOD) task, we make the first effort to establish a task-specific dataset named IOD-Bench and corresponding evaluation metrics tailored to effectively address the challenges inherent in this novel task.

**GPT-assisted Instruction Detection Data Generation.** The community has witnessed the flourishing development in the field of object detection, giving rise to numerous high-quality public datasets, such as COCO (Lin et al., 2014a) and Objects365 (Shao et al., 2019). However, there still lacks datasets specifically tailored for instruction detection, partially due to the task's ambiguous definition and the high cost of manual annotations. Inspired by the remarkable achievements of recent GPT models (Brown et al., 2020; OpenAI., 2023; 2022) in natural language understanding and reasoning, we propose utilizing the capabilities of large-scale language models (OpenAI., 2022) to generate an Instruction-oriented Object Detection dataset based on existing public object detection datasets, thereby circumventing the laborious manual annotation process.

**Instruction type 1**
1. Identify every object present in the image.
2. Spot and label all items in the scene.
3. Locate and classify all objects visible in the picture.
4. Recognize and name every item captured in the image.
5. Analyze the scene and identify each object within it.
6. …

**Instruction type 3**
1. Discover the kitchenware.
2. Gather all the stationery objects.
3. Identify all the vehicles for transportation.
4. Discover all the electronics.
5. Find all the sports accessories.
6. …

**Instruction type 2**
1. Identify the {obj} present in the image.
2. Assess the image to reveal the {obj} that may be depicted.
3. Analyze the image to find any {obj} that may be present.
4. In the image, identify {obj} that are present.
5. Look at the image and find the {obj} it contains.
6. …

**Instruction type 4**
1. Prepare a healthy salad for lunch.
2. Pack the necessary items for a day trip.
3. Create a still-life drawing.
4. Clean the spill on the kitchen floor.
5. Put out a fire quickly and efficiently.
6. …

Figure 2: Examples of constructed 4 types of instructions.

Specifically, we provide the category names from Objects365 to the LLM (OpenAI., 2022) and request it to select several class names from the given list to construct instructions while returning the target class names involved in each instruction. Since Objects365 itself contains bounding box annotations for each class, this approach enables us to obtain a dataset with both instructions and corresponding bounding box annotations for the targets. To ensure diverse instructions that cater to the varied needs of everyday scenarios, we consider 4 types of instructions:

1. **Detect All Categories**: Instructions that requires detecting all objects within an image.

2. **Detect Some Categories**: Instructions that mandate the detection of user-specified objects (similar to the current open-vocabulary object detection tasks).

3. **Detect Categories Belonging to a Super-category**: Instructions that request the detection of all subclass targets associated with a particular parent class.

4. **Detect Objects that Can Achieve Certain Goals**: Instructions that express a specific daily life requirement and demand the detection of related objects.

For generating the above four types of instructions, we devise corresponding prompts to query ChatGPT. Please refer to the Appendix for detailed prompts.

**Model Forgetting Issue.** During the generation process of the 4th type of instruction, we observed that providing ChatGPT with all 365 class names simultaneously presents a considerable challenge for the model to remember them all. This issue could cause the model to produce class names related to the instructions, which unfortunately, are not present in Objects365. In addition, it tends to overuse certain class names, thereby leading to the generation of numerous similar instructions. To alleviate the difficulty of memorizing the object name list and increase instruction diversity, we opt for sampling 30-40 class names to form an object name batch for each query process. We design a sampling strategy that makes it more likely for categories belonging to the same parent class being sampled collectively (as they are more apt to serve identical instructions). Additionally, we employ a class-balanced sampling for the class names across batches, ensuring that the generated instructions cover a balanced range of categories.

**Data Post-processing.** Despite adopting the described strategy, the generated instructions still contained categories not covered by Objects365. We filter away these extraneous categories and remove the duplicated instructions. Finally, we obtained approximately 8,000 instructions in total and their corresponding related class names across the four distinct instruction types. We allocated 80% of these instructions for training, while the remaining is reserved for evaluation.

**Dataset Construction and Partition.** To construct the dataset, for each task, we first randomly sample an object category that exists in each image, then sample an instruction that is related to the category, which establishes the image-instruction pairs. Afterwards, for each image-instruction pair, we keep all the annotations of the positive objects that match the instruction and abandon the unmatched object annotations. Since only these retained annotations are treated as the ground truth annotations for the image-instruction pairs, the AP will be decreased if objects that are irrelevant to the instructions are predicted.

To examine the generalization ability over user instructions, we adopt a rigorous approach in splitting the instructions for each task into 8:2 subsets, where 80% of the instructions are allocated as the in-domain set and the remaining 20% are reserved as the out-of-domain set. The training dataset is constructed using the in-domain instructions and training images from object365, while the validation set is constructed using 20,000 images from the validation split of Object365, paired with both in-domain and out-of-domain instructions.

**Evaluation Metrics.** During evaluation, since the LLM-generated answers may not be exactly matching to the category names in the Object365 dataset, we use BERT (Devlin et al., 2018) to extract the word embeddings of both the generated words and the ground truth categories, then calculate their similarity score and map the generated answers to ground truth by selecting the category with highest similarity. If all the scores are lower than a threshold (0.4), the prediction will be treated as false positive.

## 4 METHOD

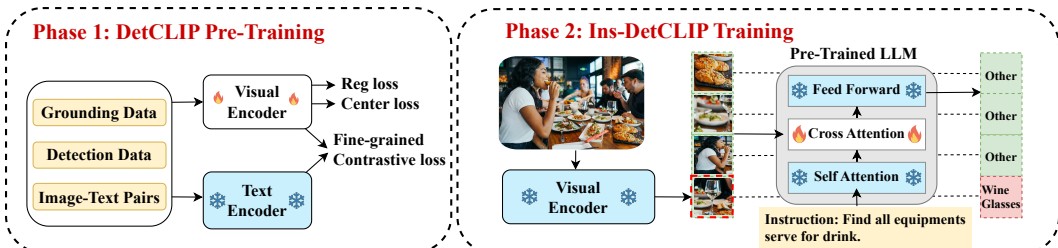

Figure 3: The overall framework of Ins-DetCLIP. The training is conducted in 2 phases. The first phase resembles the pre-training of open-vocabulary object detectors, where we adopt training data from detection, grounding and captioning tasks. In the second phase, we empower the object detector with the ability to follow human instructions by introducing large language model into the model. The model is trained on our IOD-Bench to predict object category names only for those objects that match the instruction.

Based on the proposed dataset, we present a novel object detection system named Ins-DetCLIP. Our model promotes the ability to detect any objects when provided with human language instructions. This capability is made possible by the integration of two key components: 1) the powerful open-vocabulary detectors as the visual encoder to provide fine-grained visual features; 2) the LLM with vast amount of knowledge to interpret human instruction and generate corresponding output. In this section, we will provide a detailed overview of our proposed method. We begin by outlining the design of our model and follow up with a comprehensive explanation of our training scheme.

### 4.1 MODEL DESIGN

**Visual Encoder.** The visual encoder is an open-vocabulary object detector pretrained using combined datasets from detection, grounding and captioning tasks following Yao et al. (2022b; 2023). Different from the encoders used in previous multimodal models that extracts image-level features (Zhu et al., 2023; Alayrac et al., 2022; Li et al., 2023a), the encoder of Ins-DetCLIP is not only able to extract fine-grained features at object level, but also locate the objects in the image. In addition, text-aligned object visual features facilitate instruct tuning in the later phase.

**Object-level Cross Modal Fusion.** To enable the interpretation and the ability to follow human instructions, we resort to the pre-trained large language models (LLM). Specifically, we primarily adopt the FlanT5 (Chung et al., 2022) model family as the candidates for the language decoders due to their superior instruction-following ability. We fuse visual and language features by inserting cross-attention layers among the self-attention layers of the language model's decoder, which are randomly initialized and trained from scratch. To ensure stable training at the beginning, we adopt tanh-gating mechanism following Alayrac et al. (2022), which can be formulated as follows:

$$h_{\text{L}+1} = FF[tanh(\alpha) \times XAttn(Attn(h_{\text{L}}), V) + Attn(h_{\text{L}})] \qquad (1)$$

where $FF$, $XAttn$ and $Attn$ are feed forward network, cross attention layer and self attention layer, respectively. $h_{\text{L}}$ is the hidden states at L$^{th}$ layer, and $V$ is the object visual feature. $\alpha$ is a learnable

scalar initialized to 0. In this way, the output of LLM remains unchanged in the beginning, and the visual features can be gradually integrated with the language model without disrupting the outputs.

## 4.2 TWO-PHASE TRAINING SCHEME

**Phase 1: Open-vocabulary Pretraining.** In the first phase, we conduct OVD pretraining following DetCLIP (Yao et al., 2022b). As show in Phase 1 of Fig. 3, we pre-train the detector by optimizing the fine-grained contrastive loss between text embeddings and object-level visual embeddings, along with the centerness loss and bounding box regression loss. We utilize datasets from detection, grounding and image-text pairs for conducting visual-textual feature alignment. After the first phase, we derive a powerful open-vocabulary object detector which is able to extract visual embeddings well aligned with textual embedding from the pre-trained CLIP (Radford et al., 2021) text encoder.

**Object Feature Extraction.** After the first stage, the DetCLIP model is able to propose bounding boxes given a list of category names. However, in our proposed IOD task, only open-ended user instructions are provided. To address this issue, we introduce an extra classification head that distinguishes foreground objects from the background in a class-agnostic manner, resembling region proposal network (RPN). The selected regional features are then treated as object features and provided to the LLM, which then predict the categories in a generative manner.

**Phase 2: Object-Level Instruction Tuning.** During the second phase, we leverage the training data in our IOD-Bench to perform instruction tuning. Specifically, we first freeze both the visual encoder obtained from the first stage and the pre-trained language model (Phase 2 of Fig. 3). To enable cross-modality fusion. Then, we insert randomly initialized cross attention layers within the decoder layers of the language model and train them from scratch. The image is processed through the visual encoder, which extracts object-level visual features. Simultaneously, the accompanying textual instruction is passed through the language model. Cross attention is performed between the object-level visual features and the textual features. Finally, the language modeling loss is optimized on the language model's outputs, which can be formulated as follows:

$$\mathcal{L}_{\text{instruct}} = -logp[y_t^i|\phi(y_{(<t)}^i, V^i)] \tag{2}$$

where $\phi$ denotes the LLM, $V^i$ is the object visual feature, $y_t^i$ is the textual token associated with $i^{th}$ object at $t^{th}$ time step.

**Target for Autoregressive Training.** The main philosophy behind instruct-following behavior is to enable the detector to localize only the objects of interest. We put forth a simple yet effective strategy that helps achieve this goal: In cases where the object is deemed relevant to the instruction (positive), the target output for the language model is set to be the object's category name. Conversely, if the object is deemed to be unrelated to the instruction (negative), the target output is set to "other". For every training step, we randomly sample a instruction for each object feature, and set the probability of sampling negative instructions to be a fixed constant $P_{neg}$. During inference, we can simply filter away objects that are associated with the output "other".

**Parallel Training Formulation.** After extracting object-level visual features from the image using DetCLIP, there are two ways to train the model to achieve our goal. The first approach is to concatenate the object features and train the language model to sequentially predict the output for each object. However, this approach will output an extremely long sequence if the number of objects is large. Alternatively, we can allow the language model to process each object feature independently. Specifically, each object feature interacts with the corresponding instruction via the LLM, then the LLM only predicts the output for that object. In this way, the prediction of objects can be parallelized, which promotes efficiency during inference.

## 5 EXPERIMENTS

### 5.1 IMPLEMENTATION DETAILS

**Model Architecture.** For visual encoder, we adopt the same architecture as DetCLIP (Yao et al., 2022a). Specifically, we use ATSS architecture (Zhang et al., 2020) with the same swin-T (Liu et al., 2021b) backbone. For the LLM, we mainly adopt FlanT5 model family (Chung et al., 2022). We conduct experiments on both FlanT5-base and FlanT5-Large.

Table 1: Performance comparison between our Ins-DetCLIP and two-stage counterparts on the validation set of IOD-Bench. Our method outperforms the baselines by a large margin.

| MLLM | DETECTOR | IN DOMAIN AP (TASK1 / TASK2 / TASK3 / TASK4) | OUT OF DOMAIN AP (TASK1 / TASK2 / TASK3 / TASK4) |
|---|---|---|---|
| BLIP2-FLANT5-BASE(LI ET AL., 2023A) | DETCLIP | 3.14 (4.27 / 4.15 / 2.13 / 2.02) | 3.06 (4.16 / 4.01 / 2.11 / 1.95) |
| BLIP2-FLANT5-XL(LI ET AL., 2023A) | DETCLIP | 3.95 (5.37 / 4.51 / 3.04 / 2.89) | 3.71 (5.21 / 4.40 / 2.79 / 2.43) |
| MINIGPT4-VICUNNA-7B (ZHU ET AL., 2023) | DETCLIP | 8.29 (12.3 / 10.3 / 7.51 / 3.05) | 6.35 (10.4 / 7.13 / 5.29 / 2.57) |
| MINIGPT4-VICUNNA-7B (ZHU ET AL., 2023) | GROUNDINGDINO | 8.22 (11.7 / 10.8 / 7.29 / 3.10) | 6.34 (10.6 / 7.32 / 5.04 / 2.39) |
| LLAVA-VICUNNA-7B (LIU ET AL., 2023A) | GROUNDINGDINO | 8.86 (14.5 / 11.2 / 6.46 / 3.30) | 8.91 (14.2 / 11.5 / 6.02 / 3.94) |
| INS-DETCLIP-OPT1.3B | DETCLIP | **14.9** (22.9 / 14.7 / 11.5 / 10.4) | **11.4** (20.4 / 13.6 / 7.42 / 4.10) |
| INS-DETCLIP-FLANT5-BASE | DETCLIP | **15.3** (24.5 / 15.3 / 11.3 / 10.0) | **13.7** (24.2 / 16.0 / 8.62 / 5.90) |
| INS-DETCLIP-FLANT5-LARGE | DETCLIP | **16.2** (25.6 / 16.4 / 11.7 / 11.0) | **14.4** (25.4 / 16.2 / 9.65 / 6.50) |

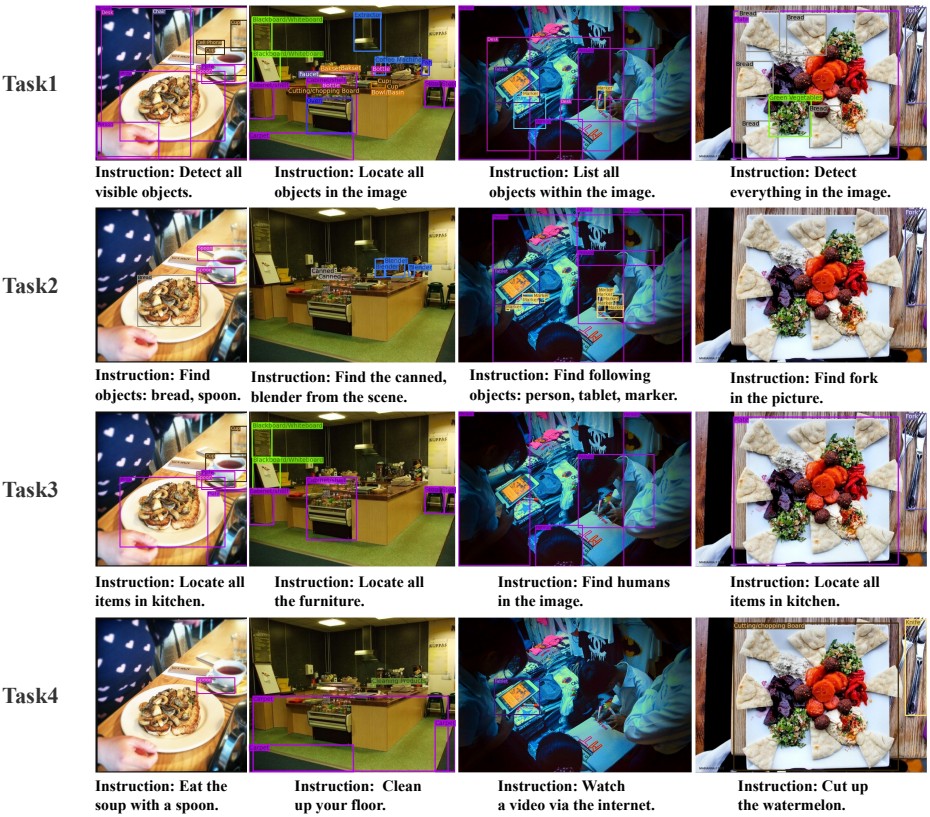

Figure 4: Visualization of bounding box results generated by our Ins-DetCLIP given different instructions. Each row from top to bottom are results of one task. The text below shows the instructions associated with the given image. We observe that our Ins-DetCLIP is able to respond differently according to different user instruction and detect the objects of interest.

**Training.** Our open-vocabulary pre-training phase follows the training settings of DetCLIP, utilizing the Objects365 and Goldg datasets. We employ a cosine decay learning rate, starting from $2.8e^{-4}$, and conduct the training over 12 epochs using 32 GPUs. For the instruction tuning phase, if not otherwise specified, we set $P_{neg}$ to 0.1, the cross attention fusion layers are inserted into the language model's decoder every $3^{rd}$ layer. The models are trained for 12 epochs. The initial learning rate is set to $2.5e^{-5}$ and is decayed by a factor of 0.1 at the 8-th and the 11-th epoch. The max token length is set to 512 following (Chung et al., 2022).

## 5.2 BASELINE MODELS

We establish baseline methods for comparison with several powerful multimodal models, such as BLIP2 (Li et al., 2023a) and MiniGPT4 (Zhu et al., 2023) variants. To achieve instruction-oriented detection, we directly integrate those models with open-vocabulary detectors (OVD). Specifically, detection is performed in a two-stage manner: we first ask the multimodal models to elicit the names of objects in the image that are relevant to the instruction, then use those names as prompts to query

the OVD model. For fair comparison, we finetune the LLM component of the baselines using the instruction-object text pairs of the in domain partition.

## 5.3 INSTRUCTION DETECTION

We demonstrate the effectiveness of our model and our training strategy by comparing with the baselines on our IOD-Bench. As shown in Table 1, our Ins-DetCLIP outperforms the counterparts by a large margin. Specifically, using FlanT5-base model, Ins-DetCLIP already outperforms MiniGPT4 baseline by 7.9 mAP averaged across all tasks. Thanks to the generalization ability of the LLMs, our model also demonstrates promising performances on instructs that do not appear during training. For instance, the Ins-DetCLIP with FlanT5-base still achieves 13.7 mAP on average for out of domain instructs, which is only slightly lower compared with results on domain instructs by 1.6 mAP.

**Discussion:** The two-stage approaches in Table 1 utilize multimodal LLMs like BLIP2, MINIGPT4 and LLAVA to identify objects, then employ open-vocabulary detectors for localization. However, MLLMs, trained with image-level features instead of object-level features, often struggle to recognize all related objects in the image. Additionally, these methods typically employ low-resolution images, impeding the recognition of small objects. In contrast, our model not only harnesses the superior capabilities of LLMs, but also utilizes the rich object-level visual features from DetCLIP, which provide more detailed information.

## 5.4 DENSE CAPTIONING RESULTS

Leveraging the exceptional generation capability of LLM, our Ins-DetCLIP can go beyond predicting only the category name, and also generate a detailed description for objects of interest. We demonstrate the superior generation ability of our model by benchmarking it on dense captioning tasks. To ensure fair comparison, we fine-tune the regression head of Ins-DetCLIP using box annotations from dense captioning datasets. As shown in Table 2, our model consistently outperforms other methods and establishes new state-of-the-art results. We also visualizaition results in the appendix.

Table 2: Performance on dense captioning benchmarks. Owing to the great generation ability of LLMs, Ins-DetCLIP achieves state-of-the-art performances on both VG 1.2 and VG COCO datasets.

| METHOD | VG V1.2 MAP(%) | METHOD | VG COCO MAP(%) |
|---|---|---|---|
| FCLN (JOHNSON ET AL., 2016) | 5.16 | FCLN | 4.23 |
| JIVC (YANG ET AL., 2017) | 9.96 | JIVC | 7.85 |
| COCD (LI ET AL., 2019) | 9.75 | COCD | 7.92 |
| CAG-NET (YIN ET AL., 2019) | 10.51 | COCG-LOCSIZ | 8.76 |
| TDC (SHAO ET AL., 2022) | 11.90 | COCG&GT | 9.79 |
| CAPDET (LONG ET AL., 2023) | 15.44 | CAPDET | 13.98 |
| GRIT (WU ET AL., 2022) | 15.50 | GRIT | - |
| INS-DETCLIP-FLANBASE | **15.70** | INS-DETCLIP-FLANBASE | **14.35** |
| INS-DETCLIP-FLANLARGE | **16.13** | INS-DETCLIP-FLANLARGE | **15.01** |

## 5.5 ABLATION STUDIES

**Cross Attention Layer Frequency.** Although it is intuitive that inserting a cross attention layer following every self attention layer achieves the best performance, it also introduces higher computational burden. Therefore, it is important that we strike a balance between the performance and the efficiency. As shown in Fig. 5, the performance drop for decreasing the frequency of cross attention layer from adding them at every layer to every $3^{th}$ layer is negligible. Therefore, we adopt this configuration for better computation efficiency.

**Inference Efficiency.** We compare the inference efficiency between Ins-DetCLIP and the 2-stage baselines in terms of frames per second (FPS) in Table 5. Our model is able to achieve better performance as well as faster inference speed compared with baseline methods.

**Size of LLM.** Even though the LLM only outputs the category name, it requires strong interpretation ability of instructions to decide which objects are relevant to user inputs. To verify this claim, in the following Table 3, we show that smaller language models have weaker interpretation ability and performs worse on our benchmark.

Table 3: We demonstrate the importance of the size of LLM. As shown in the following table, the performance of our Ins-DetCLIP increases as the size of the LLMs grow. This is because larger LLMs possess more powerful reasoning and generalization ability, which is crutial for our instruction-oriented object detection task.

| LLM | PARAMS | AVG | TASK1 | TASK2 | TASK3 | TASK4 |
|---|---|---|---|---|---|---|
| FLANT5-SMALL | 60M | 13.3 | 23.9 | 12.1 | 9.45 | 7.62 |
| FLANT5-BASE | 250M | 15.3 | 24.5 | 15.3 | 11.3 | 10.0 |
| FLANT5-LARGE | 780M | **16.2** | **25.6** | **16.4** | **11.7** | **11.0** |

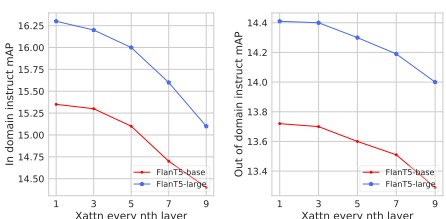

Figure 5: Performance of Ins-DetCLIP with different cross attention layer frequencies. Frequency of every $3^{rd}$ layer achieves a good trade off between computation annd performance.

**Release/freeze LLM and Detector.** In Table 4, we compare the results between Ins-DetCLIP trained with frozen or released LLM. We observe that although releasing the LLM during training slightly boosts the performance on in domain instructions, the ability to generalize to out of domain instructs is degraded. This is reasonable since tuning the entire LLM makes the model more biased towards the training data, which in turn causes loss in its original knowledge. Therefore, we keep LLM frozen during the second phase to promote generalization ability and training efficiency.

Table 4: Experiment on releasing the LLM during instruction tuning. Although in domain performance is slightly improved, out of domain performance is severely degraded.

Table 5: Inference speed comparison between Ins-DetCLIP and baselines.

| RELEASE LLM | IN DOMAIN AP (T1 / T2 / T3 / T4) | OUT OF DOMAIN AP (T1 / T2 / T3 / T4) |
|---|---|---|
| ✗ | 15.3 (24.5 / 15.3 / 11.3 / 10.0) | **13.7** (24.2 / 16.0 / 8.62 / 5.90) |
| ✓ | **15.8** (26.1 / 15.2 / 11.7 / 10.3) | 11.5 (23.8 / 10.8 / 5.76 / 5.50) |

| METHOD | INFERENCE SPEED (FPS) |
|---|---|
| BLIP2-FLANXL | 1.0 |
| BLIP2-FLANXXL | 0.8 |
| MINIGPT4-VICUNNA-7B | 0.2 |
| INS-DETCLIP-FLANBASE | **2.2** |
| INS-DETCLIP-FLANLARGE | **1.6** |

**Ratio of Negative Instructions**. In this section, we ablate on the probability of sampling negative instructions $P_{neg}$ during training. We find that if the negative ratio is too high, the model will have the tendency to output "other". On the other hand, if ratio is too low, the model will output the object's category name regardless of the user instruct. We search across several values and find that 0.1 to be the most suitable ratio.

Table 6: Experiments with different ratios of negative instructions sampled during training.

| | NEG RATIO | IN domain AP (TASK1 / TASK2 / TASK3 / TASK4) | OUT OF DOMAIN AP (TASK1 / TASK2 / TASK3 / TASK4) |
|---|---|---|---|
| FLANT5-base | 0.05 | 14.9 (25.3 / 14.6 / 10.4 / 9.52) | 13.5 (25.0 / 15.1 / 8.20 / 5.65) |
| | 0.1 | **15.3** (24.5 / 15.3 / 11.3 / 10.0) | **13.7** (24.2 / 16.0 / 8.62 / 5.90) |
| | 0.5 | 14.0 (22.3 / 14.9 / 9.97 / 8.65) | 12.2 (21.9 / 14.1 / 7.95 / 5.04) |
| FLANT5-large | 0.05 | 15.6 (25.9 / 15.9 / 11.3/ 10.4) | 14.3 (25.3 / 16.0 / 9.47 / 6.65) |
| | 0.1 | **16.2** (25.6 / 16.4 / 11.7/ 11.0) | **14.4** (25.4 / 16.2 / 9.65 / 6.50) |
| | 0.5 | 15.1 (22.9 / 15.6 / 11.4/ 10.5) | 13.3 (22.3 / 15.7 / 9.11 / 6.13) |

# 6  CONCLUSION

In this paper, we introduce a novel task of modern object detection system called Instructional Object Detection (IOD). Instead of just recognizing objects of given category lists, IOD requires the detector to understand users' instructions and finds relevant objects, bringing a fresh human-computer interaction angle to the object detection system. We develop an extensive instruction-guided detection dataset called IOD-Bench, enriched with over 8,000 diverse instructions. Moreover, based on the proposed IOD-Bench, we further propose the Ins-DetCLIP to integrate the advantages of extensive language models within the detection framework to understand human instructions and infer corresponding objects. Experiments results demonstrate the effectiveness of the proposed method on IOD-Bench. In addition, Ins-DetCLIP also achieves SOTA performance on dense captioning tasks. We hope our work can inspire further innovation in the area of language-guided detection system.

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

## A  LIMITATIONS

Although Ins-DetCLIP trained on IOD-Bench has shown promising results as the first attempt towards instruction object detection, there are still minor issues that need to be addressed. One limitation is that the category names used to construct IOD-Bench may not always align with the content of the image, resulting in inappropriate instructions being assigned to images. For example, an image of a cup in a bathroom may be assigned the instruction "I wish to drink water." To mitigate this issue, one potential solution is to use the data-generation pipeline proposed by LLAVA (Liu et al., 2023a), which leverages image captions to provide visual context to ChatGPT (OpenAI., 2022) and generate instructions that are more closely related to the image. However, this pipeline may face challenges when applied to large-scale datasets with bounding box annotations that lack caption annotations, limiting its applicability beyond COCO (Lin et al., 2014b). We will further improve our IOD-Bench in the future work to eliminate this issue.

## B  MORE IMPLEMENTATION DETAILS

**Addressing the Issue of Absent Candidate Categories.** Traditional open-vocabulary detectors require a list of object names/phrases to derive confidence score. Therefore, without prior category knowledge, one issue that arises in Ins-DetCLIP is the lack of measurement to select the bounding boxes to be feed into LLM. To address this issue, we introduce an extra classification head that distinguishes foreground objects from the background, resembling region proposal network (RPN). The selected regional features are then treated as object features and sent to the LLM for category prediction. After obtaining the category names from the language model, we calculate the class-specific scores between the predicted categories and the regional features using the text and visual encoders of DetCLIP, whose parameters are fixed during the object-level instruction tuning phase.

**Additional Hyperparameters and Details.** We conduct two non-maximum-suppression (NMS) operations during inference, similar to 2-stage object detectors. We first apply class-agnostic NMS using the scores from foreground-background classification head, where the IoU threshold is set to 0.6. We set the maximum number of region proposals from foreground-background classification head to 100, which we find to make a good trade-off between accuracy and inference efficiency. Then, we extract the text embeddings of the predicted object categories from the frozen text encoder. Subsequently, we perform class-wise NMS using the class-wise similarity scores calculated between the text embeddings and the object visual embeddings, with the IoU threshold set to 0.5. For the first stage, the training of DetCLIP-T with Swin-T backbone takes 63 hours on 32 V100 GPUs. The resulting model from the pretraining stage can already serve as powerful open-vocabulary object detector. In the second stage, it takes only around 24 hours on 16 V100 GPUs for instruction tuning.

### B.1  ADDITIONAL EXPERIMENTS

**Inference Speed Comparison with Similar Model Sizes.** We compare the inference speed of our Ins-DetCLIP with the two-stage baseline in Table 7. We observe that even if the two-stage baseline

leverages LLM with equal or smaller size than our Ins-DetCLIP, its inference speed is still slower. This is because those methods need to output the object category names sequentially, while our model is able to make predictions for different objects in parallel.

Table 7: Inference speed comparison between Ins-DetCLIP and baselines with fewer parameters. Ins-DetCLIP still demonstrates higher efficientcy.

| METHOD | INFERENCE SPEED (FPS) |
|---|---|
| BLIP2-FLANBASE | 1.4 |
| INS-DETCLIP-FLANBASE | **2.2** |
| INS-DETCLIP-FLANLARGE | **1.6** |

**Impact of First-stage Pretraining.** We compare the performance of Ins-DetCLIP w/o first stage pretraining in Table 9. In this experiment, we still utilize the ImageNet-pretrained swin-transformer weights to initialize the vision backbone, and use FILIP-prtrained text encoder. We notice that the performance of Ins-DetCLIP greatly drops if first stage pretraining is skipped. This verifies that pretraining is an essential step to ensure that the detecto is able to provide high-quality object features to the LLM.

---

**Task1:**
Devise various expressions to convey the meaning of "detecting all objects in an image". here are some examples: 1. Identify every objects within the picture. 2. Ascertain all items present in the image. Give me 10 results.

**Task2:**
Express "Detect objects in the image" using various sentence structures, where is a collection of object names and used as a placeholder. Here are some simple examples: 1. Identify the objects present in the image. 2. Perform object recognition to detect the objects in the picture. Give me 10 results.

**Task3:**
Group the following object names into different parent categories: class names of Objects365. Format each result as a instruction-objects pair as following: find all/locate/identify/discover/detect (and so on) parent category, objects: ['object1', 'object2', ...]. An example is : Look for all electronic devices: ['Moniter/TV', 'Laptop', 'Cell Phone', 'Camera', 'Computer Box', 'Tablet', 'Keyboard', 'Mouse', 'Printer', 'Projector', 'Telephone', 'Head Phone', 'Remote', 'Microphone', 'Speakers', 'Surveillance Camera', 'Air Conditioner', 'Fan', 'Router/modem']. Return all the objects belonging to the parent category. give me 50 results.

**Task4:**
Give some instructions delivering specific purpose that require the use of the following objects:[an object name list...]. To be effective, instructions should use a language that is natural and familiar, much like the way people give directions to robots. The purpose and sentence structure of the instructions should be diverse to accommodate different scenarios. For each instruction, return the related objects. Format the results as following: Instruction: XXXX, Objects: ['object1', 'object2', ...]. Only use the object names provided in the list, do not use other object names. Now, give me 10 examples."

---

Figure 6: Detailed prompts for querying 4 types of instructions.

## C MORE DETAILS FOR DATA GENERATION

**The Creation of IOD-Bench** To enable the object detection system to flexibly perform different tasks given natural language instructions, we design <instruction>-<objects of interest> pairs to help training and evaluation. Specifically, we design such pairs for each task by leveraging the generative power of ChatGPT as follows:

Table 8: The impact of first stage pretraining. We observe drastic performance degradation without the first stage pretrainng. This verifies the importance of pretraining stage, which enables the object detector to extract high-quality object features.

| MODEL | PRETRAIN | IN DOMAIN AP (TASK1 / TASK2 / TASK3 / TASK4) | OUT OF DOMAIN AP (TASK1 / TASK2 / TASK3 / TASK4) |
|---|---|---|---|
| INS-DETCLIP-FLANT5-BASE | N | **8.46** (12.6 / 10.5 / 7.16 / 3.53) | **7.75** (11.3 / 10.3 / 6.10 / 3.31) |
| INS-DETCLIP-FLANT5-BASE | Y | **15.3** (24.5 / 15.3 / 11.3 / 10.0) | **13.7** (24.2 / 16.0 / 8.62 / 5.90) |

Table 9: Comparison with directly using open-vocabulary object detectors on IOD-Bench. It can be observed that OVOD methods achieves poor performances, since they are not able to interpret natural language instructions.

| MODEL | IN DOMAIN AP (TASK1 / TASK2 / TASK3 / TASK4) | OUT OF DOMAIN AP (TASK1 / TASK2 / TASK3 / TASK4) |
|---|---|---|
| DETCLIP | **2.81** (5.35 / 4.33 / 1.29 / 0.28) | **2.97** (5.49 / 4.96 /1.10 / 0.31) |
| GROUNDINGDINO | **4.08** (4.49 / 9.33 / 2.29 / 0.21) | **3.96** (4.26 / 8.97 / 2.41 / 0.19) |
| INS-DETCLIP-FLANT5-BASE | **15.3** (24.5 / 15.3 / 11.3 / 10.0) | **13.7** (24.2 / 16.0 / 8.62 / 5.90) |
| INS-DETCLIP-FLANT5-LARGE | **16.2** (25.6 / 16.4 / 11.7 / 11.0) | **14.4** (25.4 / 16.2 / 9.65 / 6.50) |

- For task 1, we let ChatGPT produce instructions that have the same meaning as "detecting all objects in an image". In this case, all the objects that exists in the image are considered as objects of interest;

- For task 2, we let ChatGPT generate more templates similar to "detect <obj_names> in the image", where <obj_names> is a placeholder for a list of objects and are considered as objects of interest;

- For task 3, we generate the instruction-objects pairs such as "Look for all electronic devices: ['Moniter/TV', 'Laptop', 'Cell Phone', 'Camera', 'Computer Box', 'Tablet', 'Keyboard', 'Mouse', 'Printer', 'Projector', 'Telephone', 'Head Phone', 'Remote', 'Microphone', 'Speakers', 'Surveillance Camera', 'Air Conditioner', 'Fan', 'Router/modem']". The objects mentioned in the list will be considered as objects of interest for given the instruction.

- For task 4, ChatGPT needs to design high-level user instructions and select the objects of interest from the ground truth categories.

With the collected <instruction>-<objects of interest> pairs, we are able to supplement existing datasets with natural language instruction, and curate the IOD-Bench to facilitate the research in instruction-oriented object detection.

**Precise Definition of Task 4** In the context of our problem definition, the term "relatedness" refers to objects that are relevant or connected to the purpose described in the instruction. The related objects should be ones that can be used or involved in accomplishing the specific purpose outlined in the instruction. For example, if the purpose of an instruction is to assemble a piece of furniture, related objects could include tools, screws, and various components needed for assembly. Similarly, if the purpose is to prepare a meal, related objects could include ingredients, utensils, and cooking appliances. In our constructed dataset, the "related" objects of an instruction are comprehensive, which enables our model to produce all the objects that may be associated with the instruction. Since instruction-oriented detection is an intermediate step for building an intelligent system, in actual deployment, the predicted objects may be passed to another downstream module to decide what objects to be used exactly, and in which order they should be used.

**Prompts for Querying Instructions.** Fig. 6 provides the detailed prompts used for querying four types of instructions. These prompts are provided to ChatGPT to generate instruction annotations.

**Dataset Statistics.** In Table 10, we provide the detailed number of instructions for 4 proposed tasks. We generate more instructions for task 3 (692) and task 4 (7768), and less instructions for task 1 (30) and task 2 (111), since there exists more diverse instructions for super-class-related and goal-oriented tasks.

Table 10: Numbers of instructions for 4 proposed tasks.

| Task1 | Task2 | Task3 | Task4 | Total |
|-------|-------|-------|-------|-------|
| 30 | 111 | 692 | 7768 | 8601 |

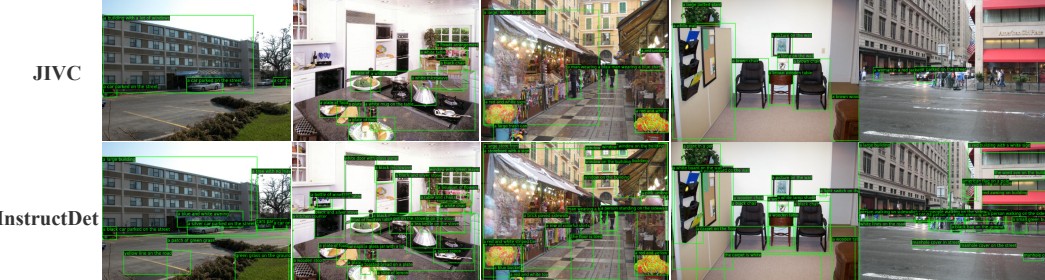

Figure 7: Qualitative visualization of JIVC and our Ins-DetCLIP on dense captioning task.

**More Visualizations.** Fig. 9 provides more visualization examples for the constructed 4 types of instructions in our curated IOD-Bench. It demonstrates a well correspondence between the instructions and the objects illustrated in the images.

## D  DEMONSTRATIONS

**Instruction-oriented Object Detection.** In this section, to showcase the superior instruction-following ability of our Ins-DetCLIP, we demonstrate more examples obtained with Ins-DetCLIP for each task. As shown in Fig. 8, Fig. 10, Fig. 11 and Fig. 12, our Ins-DetCLIP is able to flexibly make predictions based on user instructions and locate objects of interest in the image.

**Dense Captioning** We visualize the dense captioning results in Figure 7, which demonstrates that Ins-DetCLIP is able to provide detailed descriptions for each object in the image.

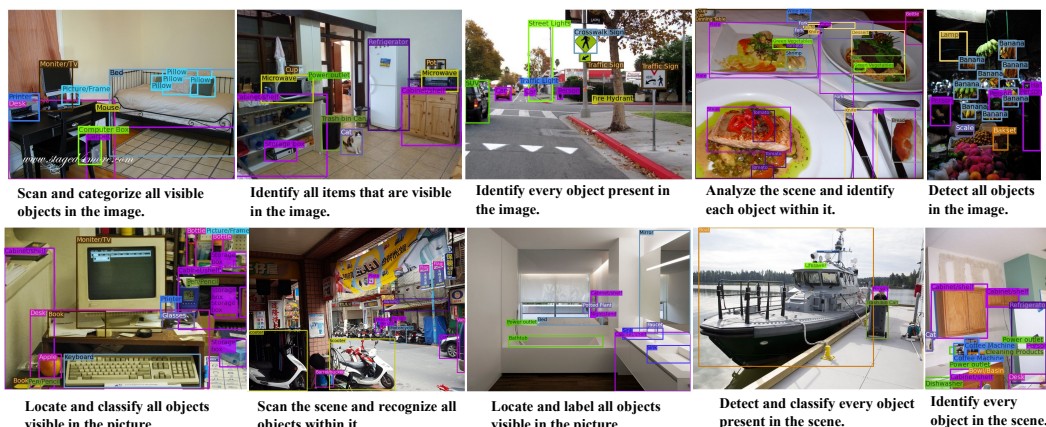

Figure 8: Visualization of bounding box results generated by our Ins-DetCLIP for Task 1, which detects all the visible objects in the image.

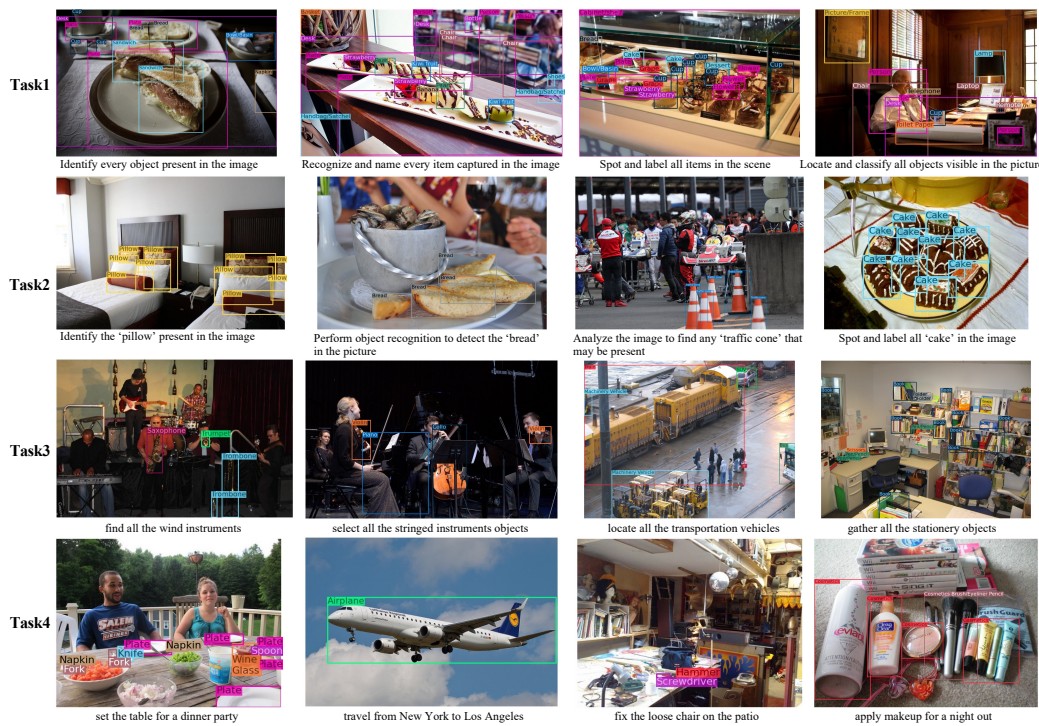

Figure 9: Examples of the constructed 4 types of tasks and their corresponding instructions in IOD-Bench. The instructions align well with the corresponding object categories.

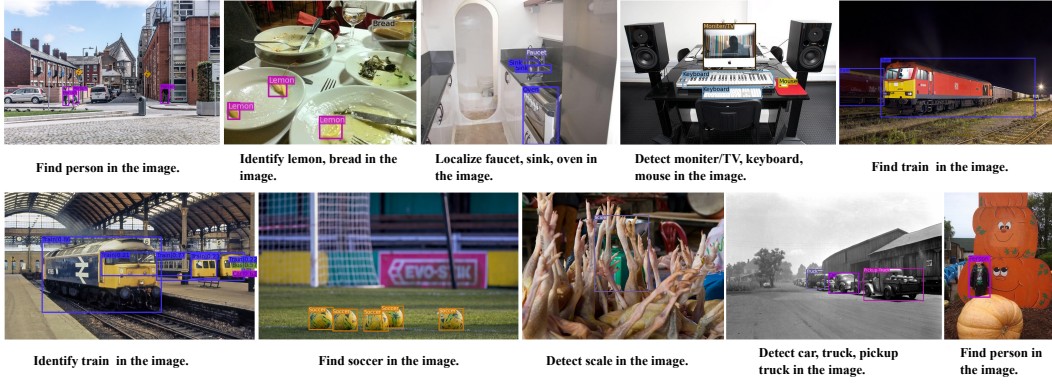

Figure 10: Visualization of bounding box results generated by our Ins-DetCLIP for Task 2, which detects specific object categories in the image.

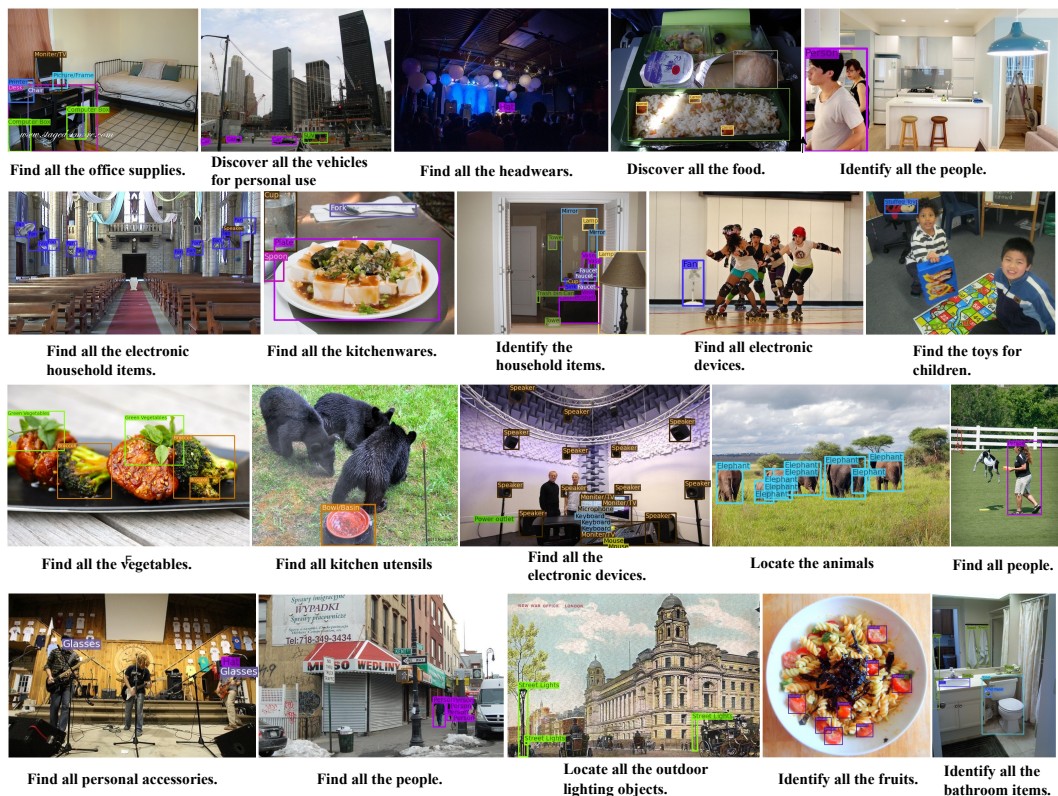

Figure 11: Visualization of bounding box results generated by our Ins-DetCLIP for Task 3, which detects objects that falls under a super-category.

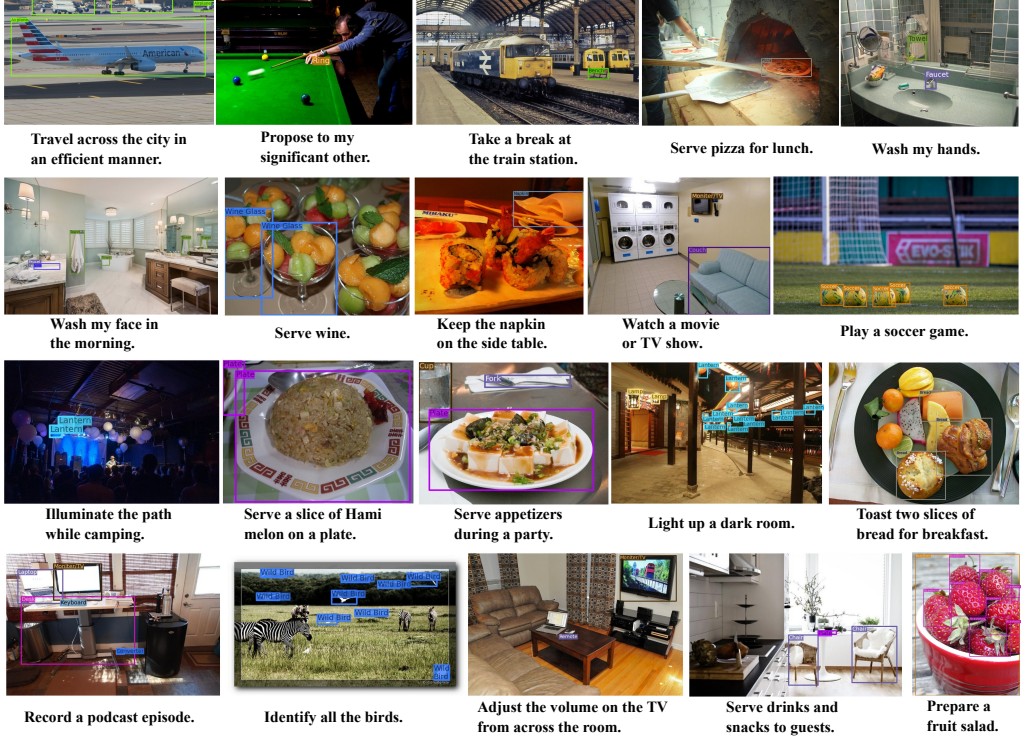

Figure 12: Visualization of bounding box results generated by our Ins-DetCLIP for Task 4, which detects objects that are helpful for fulfilling a goal.

