# OpenReview forum: "Ins-DetCLIP: Aligning Detection Model to Follow Human-Language Instruction"
_ICLR.cc/2024/Conference — ICLR 2024 poster_

### Official Review · Reviewer_t19r · 2023-10-31

**Soundness:** 2 fair
**Presentation:** 3 good
**Contribution:** 2 fair
**Rating:** 5
**Confidence:** 4

**Summary:**

This paper proposes a novel detection task, called IOD, which requires the detector to detect objects related to the human instruction. To unravel the problem, the authors first created a dataset called IOD-Bench, and then designed an Ins-DetCLIP model. The idea is simple and straightforward -- by incorporating an LLM with an open-vocabulary detector. Experiments on the IOD-Bench show that Ins-DetCLIP outperforms the baseline methods.

**Strengths:**

- The proposed IOD task is novel and has practical values.
- The method is simple and easy to understand.

**Weaknesses:**

- Instruction types 1, 2, and 3 seem to be variants of traditional detection or OV detection. With the help of LLM, it should be easy to convert these instructions to traditional detection or OV detection problems. Intuitively, I believe that using off-the-shelf detectors and LLM will achieve good results on these types of instructions. If not, a discussion of the reason is expected.
- Instruction type 4 is abstract enough to meet my expectations of IOD. However, the objective of instruction type 4 is to detect related objects, which is vague. For instance, should cutleries be detected if the instruction is "Prepare a healthy salad for lunch"?
- In Phase 2 of Ins-DetCLIP training, the visual encoder is frozen. This suggests that the LLM can only pick objects from the object proposals given by DetCLIP. However, DetCLIP is unaware of the instructions. As the instruction becomes more complex, DetCLIP will be unable to recall the target objects with a fixed number of proposals.
- There have been several multimodal LLMs that are capable of detecting objects, i.e. Shikra[1], Kosmos[2]. The proposed Ins-DetCLIP should be compared with such methods.

**Questions:**

- As for the evaluation metrics, the authors said that they use BERT to compute the similarities between the predicted categories and the GT categories. What if the prediction is completely irrelevant to the GT categories? In this case, the similarity score distribution will likely to be random. Will this cause the metrics to be unstable?

---

> ### Author Response · Authors · 2023-11-17
> **Response to Reviewer t19r (1)**
>
> We sincerely thank Reviewer t19r for your review and are grateful for the time you spent on our submission. We are also glad you think our proposed IOD task is novel and has practical values. Below we would like to give detailed responses to each of your comments.
>
> **1. Instruction types 1, 2, and 3 seem … using off-the-shelf detectors and LLM will achieve good results.**
>
> Thanks for your question. Even though instruction types 1, 2, and 3 look like traditional detection tasks, they are fundamentally different: our task involves identifying items based on any free-form instruction with unlimited categories, as opposed to traditional detection methods, which are limited to a fixed number of classification heads.  We also prevent the need for the category names as in open-vocabulary detection. Due to the nature of our tasks, using off-the-shelf detectors and LLM is hard to convert these instructions tasks to traditional detection. We wish to clarify from two aspects:
>
> **(1) Task 1,2 and 3 are not merely traditional detection tasks.**
>
> We wish to first recall the definition of the first 3 tasks:
> - task 1 requires  identifying all visible objects;
> - task 2 requires identifying certain objects given their category names;
> - task 3 needs to discover objects that fall under a parent category and predict their exact categories.
>
> We wish to note that although task 1 and task 3 look like traditional detection tasks, e.g., detecting all 80 categories on COCO dataset, they are fundamentally different, since our tasks are open-world, rather than limited to a fixed number of classes as in traditional detection tasks. Meanwhile, Task 2 requires detecting specific objects given their categories, which is similar to open-vocabulary object detection and is also a frequent case of instruction.
>
> **(2) Off-the-shelf LLM+OV detector can not adequately address IOD task**
>
> Combining only the off-the-shelf  LLM with open-vocabulary detector for IOD task would be either one of the two approaches: 1) the detector first discovers all the objects in the image given a list of categories, then the LLM selects the objects that match the user's instruction; 2) the LLM first selects the object categories that are relevant to the instruction, then the detector discovers those objects based on the selected categories. However, the first approach requires a list of object names that is comprehensive enough to cover all the objects that may appear, which is impractical;  In the second approach, without knowing what objects might exist in the image, it is infeasible for the LLM to select objects only given the user instruction.
>
> However, your suggestion about using off-the-shelf detectors and LLM raises a very good point. Our 2-stage baseline approach is also inspired from this view, which prevents the need for the category list by leveraging the multimodal LLM to list the related object names based on the input image and instruction. The experimental results of the 2-stage baselines are shown in Table 1 of the main paper. We copy the table below for reference:
>
> | MLLM                           | Detector        | In domain                                      | Out of Domain                                   |
> |--------------------------------|-----------------|------------------------------------------------|------------------------------------------------|
> | MiniGPT4-Vicunna-7B            | DetCLIP         | 8.29 (12.3 / 10.3 / 7.51 / 3.05)               | 6.35 (10.4 / 7.13 / 5.29 / 2.57)               |
> | MiniGPT4-Vicunna-7B            | GroundingDino   | 8.22 (11.7 / 10.8 / 7.29 / 3.10)               | 6.34 (10.6 / 7.32 / 5.04 / 2.39)               |
> | LLAVA-Vicunna-7B               | GroundingDino   | 8.86 (14.5 / 11.2 / 6.46 / 3.30)               | 8.91 (14.2 / 11.5 / 6.02 / 3.94)               |
> | Ins-DetCLIP-FlanT5-base        | DetCLIP         | **15.3** (24.5 / 15.3 / 11.3 / 10.0)           | **13.7** (24.2 / 16.0 / 8.62 / 5.90)           |
> | Ins-DetCLIP-FlanT5-large       | DetCLIP         | **16.2** (25.6 / 16.4 / 11.7 / 11.0)           | **14.4** (25.4 / 16.2 / 9.65 / 6.50)           |
>
>
> The two-stage baselines demonstrate inferior performances due to the following drawbacks:
> - input images of the VIT encoder typically have low resolution, which makes them unable to identify small objects;
> - the MLLM are known to suffer from hallucination, which often mention objects that do not appear in the image;
> - the LLM encounter difficulty in naming all the relevant objects sequentially when the number of objects grows large.
>
> On the other hand, our approach demonstrates advantage in the following aspects: 1) we adopt DetCLIP to extract fine-grained object level features from high-resolution images, which prevents  the issue with small objects;  2) the LLM directly performs reasoning directly for each object feature in parallel, which reduces the difficulty for generation, achieves better results and boosts the efficiency.

---

> ### Author Response · Authors · 2023-11-17
> **Response to Reviewer t19r (2)**
>
> **2. The objective of instruction type 4 is vague.**
>
> Thank you for pointing out this potential confusion. In the context of our problem definition, the term "relatedness" refers to objects that are relevant or connected to the purpose described in the instruction. The related objects should be ones that can be used or involved in accomplishing the specific purpose outlined in the instruction. For example, if the purpose of an instruction is to assemble a piece of furniture, related objects could include tools, screws, and various components needed for assembly. Similarly, if the purpose is to prepare a meal, related objects could include ingredients, utensils, and cooking appliances. Therefore, in your question, the cutleries should be considered as "related objects" to the instruction "prepare healthy salad for lunch".
>
> In our constructed dataset, the "related" objects of an instruction are comprehensive, which enables our model to produce all the objects that may be associated with the instruction. Since instruction-oriented detection is an intermediate step for building an intelligent system, in actual deployment, the predicted objects may be passed to another downstream module to decide what objects to be used exactly, and in which order they should be used. We have updated our paper in Section C of the Appendix and hope to better clarify this issue.
>
> **3. The visual encoder is frozen. This suggests that the LLM can only pick objects from the object proposals given by DetCLIP.**
>
> Thank you for the question. During fine-tuning, we enable the DetCLIP to output object features in a class-agnostic manner by training a lightweight classification head to distinguish the foreground proposals from the background (elaborated in Section 4.2 of the updated paper). This is to ensure that the features produced by DetCLIP can encompass the majority of the visible objects, as comprehensively as  possible. Afterwards, the reasoning and decision-making processes are conducted by the LLM. In this way, DetCLIP behaves like an instruction-agnostic visual feature extractor on the object-level, which is analogous to the VIT-based image-level visual feature extractors used in multi-modal large language models, such as MiniGPT-4 and LLAVA. However, we agree that enabling different visual features to be extracted given various instructions could be an interesting direction to explore, which may potentially lead to higher performance. InstructBLIP (cited in related work of the updated version) has made attempts to achieve this for image-level multimodal understanding. We will leave this exploration to future work.
>
> **4. Comparison with Shikra and Kosmos2.**
>
> Thank you very much for the suggestion. We first update these methods in the related work section of our paper. Then, we perform extra experiments to compare with the mentioned approaches on our benchmark, and the results are demonstrated below (updated in Table 10 of the Appendix).
>
> Even though these approaches excel at tasks such as referring expression comprehension or spot caption, we observe that such methods demonstrate lower performances on our benchmark, which may be attributed to the following two reasons:
> - these methods adopt low resolution input images due to the VIT image encoders, which makes the small objects difficult to be recognized;
> - the sequence representation of bounding boxes make it difficult for the LLM to output all the coordinate locations when there are many target object related to the instruction.
>
> However, we believe these approaches that integrate the fine-grained perception abilities into LLMs are promising directions, which deserve to be explored more in the future.
>
> | Model                     | In domain                                      | Out of Domain                                   |
> |---------------------------|------------------------------------------------|------------------------------------------------|
> | Kosmos-2 [1]                  | **7.51** (11.5 / 9.61 / 6.30 / 2.64)           | **6.98** (11.0 / 9.13 / 5.30 / 2.47)           |
> | Shikra [2]                    | **7.10** (9.57 / 9.20 / 6.43 / 3.22)           | **6.82** (9.10 / 8.93 / 6.21 / 3.04)           |
> | Ins-DetCLIP-FlanT5-base    | **15.3** (24.5 / 15.3 / 11.3 / 10.0)           | **13.7** (24.2 / 16.0 / 8.62 / 5.90)           |
> | Ins-DetCLIP-FlanT5-large   | **16.2** (25.6 / 16.4 / 11.7 / 11.0)           | **14.4** (25.4 / 16.2 / 9.65 / 6.50)

---

> ### Author Response · Authors · 2023-11-17
> **Response to Reviewer t19r (3)**
>
> **5. As for the evaluation metrics, What if the prediction is completely irrelevant to the GT categories?**
>
> Thank you for this question. We wish to note that the similarity score calculated using BERT is leveraged to map the model's prediction to the ground truth category of the dataset. If the score is lower than a preset threshold (in our case, we  set it  to be 0.4), the prediction is going to be treated as false positive and decrease the mAP score. We update this missing information in our paper. Afterwards, we follow the conventional evaluation pipeline for object detection. Since we follow the same evaluation protocol for all the methods, the results are comparable and are able to reflect the true performance.
>
> However, we agree that there is no standard and well established evaluation metric for generation-based tasks. For example, many recent works use powerful LLMs such as GPT-4 to evaluate the correctness of the generated answers. Unfortunately, such evaluation may not be scalable, especially in the case of IOD tasks, where the number of object predictions that need to be evaluated is huge. How to properly conduct evaluation for such tasks is indeed a research topic that has profound impact. We will leave its investigation in our future work.
>
> Overall, we sincerely thank you for your insightful points and your time spend on our paper. We have added the experiments and revised statements accordingly in our current version. We hope our answers have addressed each of your concerns. If you have any further questions, we are happy to address them.
>
> **References:**
>
> [1] Kosmos-2: Grounding Multimodal Large Language Models to the World
>
> [2] Shikra: Unleashing Multimodal LLM’s Referential Dialogue Magic

---

> ### Author Response · Authors · 2023-11-21
> **Waiting for further discussion.**
>
> Dear Reviewer t19r,
>
> We are grateful for your valuable time during the reviewing process, and we appreciate all your constructive comments on our work. We are also very glad that you acknowledge that our proposed IOD task is novel and has practical values, and the method is simple and easy to understand.
>
> Since it has come to the end of the first rebuttal phase, we sincerely hope that our response has addressed your problems and will continue to address further questions if you have any.
>
> Thank you!

---

> ### Author Response · Authors · 2023-11-23
> **Waiting for further discussion.**
>
> Dear Reveiwer t19r,
>
> We really appreciate your insightful comments and your acknowledgement that our proposed task is novel. We have updated our draft and added replies with our latest experimental results. In previous response,
>
> - We provided justification with experiments for why using off-the-shelf detectors and LLM cannot achieve good results.
> - We add a detailed definition and explanation about the objective of task 4.
> - We conduct additional experiments to compare Shikra and Kosmos-2 on IOD task.
> - We discussed about the effect of frozen visual encoder.
>
> Since the rebuttal deadline is approaching this evening, a lot of papers have finished the discussion. Given that your current score is 5, we would appreciate it if you could let us know if our responses have addressed your concerns satisfactorily. If your concerns have not been resolved, could you please let us know about it so that we have the opportunity to respond before the deadline?
>
> We understand you are very busy and we really appreciate your time. We would be happy to have any follow-up discussions or address any additional concerns.
>
> Thanks very much! Looking forward to your reply.
>
> Paper4489 Autors

---

### Official Review · Reviewer_Y4Gg · 2023-10-31

**Soundness:** 3 good
**Presentation:** 3 good
**Contribution:** 3 good
**Rating:** 5
**Confidence:** 3

**Summary:**

In this paper, the authors introduce a novel task named Instruction-oriented Object Detection (IOD). In the course of developing an IOD system, an innovative dataset, IOD-Bench, is presented, along with a proposed model named Ins-DetCLIP.

**Strengths:**

The idea that the author solves real-world problems through diverse instructions is interesting.

**Weaknesses:**

1-I understand the authors' aspiration to address a wide range of real-world scenarios through diverse instructions. However, I find the motivations behind the four tasks to be somewhat unclear. The similarity amongst the first three tasks appears quite pronounced, and it seems that existing datasets already cater to these scenarios to a certain extent.

2-In Section 5.2, the authors described integrating BLIP2 and MiniGPT4 with open-vocabulary detectors (OVD) in a two-stage sequential manner to construct baseline models. I'm curious as to why such a sequential approach was chosen for building the baseline models. Would it not be more straightforward to employ the OVD method for direct open-vocabulary detection? My concern is that a two-stage sequential manner could inherently limit the system's performance to the accuracy of both models, seeming like a suboptimal choice.

**Questions:**

see Weaknesses

---

> ### Author Response · Authors · 2023-11-17
> **Response to Reviewer Y4Gg (1)**
>
> We sincerely thank Reviewer Y4Gg for the review and are grateful for the time you spent with our submission. We are glad for the acknowledgement that our proposed idea is interesting, and we wish to address your concerns by giving detailed responses to each of your comments as follows:
>
> **1. The similarity amongst the first three tasks appears quite pronounced, existing datasets already cater to these scenarios.**
>
> Thanks for pointing out this potential confusion. We wish to clarify that even though the first three tasks may look similar, they are fundamentally different, which would expect different instructions from the user,  and the existing datasets alone do not cater for our needs.
>
> We wish to recall the definition for each task first:
> - task 1 requires  identifying all visible objects;
> - task 2 requires identifying certain objects given their category names;
> - task 3 needs to discover objects that fall under a parent category and predict their exact categories.
>
> To enable the object detection system to flexibly perform such tasks given natural language instructions,  we not only need the existing datasets containing the box annotations for objects, but also the natural language instructions that indicate which task to complete. Therefore, the key is to design <instruction>-<objects of interest> pairs to help training and evaluation. Specifically, we design such pairs for each task by leveraging the generative power of ChatGPT as follows:
>
>
> * for task 1, we let ChatGPT produce instructions that have the same meaning as “detecting all objects in an image”. In this case, all the objects that exists in the image are considered as objects of interest;
> * for task 2, we let ChatGPT generate more templates similar to “detect <obj_names> in the image”, where <obj_names> is a placeholder for a list of objects and are considered as objects of interest;
> * for task 3, we generate the instruction-objects pairs such as “Look for all electronic devices: [’Moniter/TV’, ’Laptop’, ’Cell Phone’, ’Camera’, ’Computer Box’, ’Tablet’, ’Keyboard’, ’Mouse’, ’Printer’, ’Projector’, ’Telephone’, ’Head Phone’, ’Remote’, ’Microphone’, ’Speakers’, ’Surveillance Camera’, ’Air Conditioner’, ’Fan’, ’Router/modem’]. The objects mentioned in the list will be considered as objects of interest for given the instruction.
>
> Combining the collected <instruction>-<objects of interest> pairs and the bounding box annotations of existing detection datasets, we are able to curate the IOD-Bench to facilitate the research in instruction-oriented object detection. We greatly appreciate the advice and update the detailed motivation and explanation for the construction of IOD-Bench in Section C of the Appendix.

---

> ### Author Response · Authors · 2023-11-17
> **Response to Reviewer Y4Gg (2)**
>
> **2. Why not use OVD method for direct open-vocabulary detection?**
>
> Thank you for the question. We agree that employing the OVD method for IOD tasks is the most straight-forward baseline to think of. However, we wish to clarify that open-vocabulary detectors can only detect objects given the exact category names, and do not have the capability to detect objects of interest given an open-ended user instruction. For instance, to detect the 80 categories of COCO, those category names need to be directly provided to the detector. The detector can not work with language instructions such as "detect all visible objects", "detect all the sports equipment", or "detect a place where I can rest comfortably", since the object names are not directly given to the detector.
>
> To verify this, we directly use open-vocabulary object detectors for the four tasks, and demonstrate the results as follows:
>
> | Model                     | In domain                                      | Out of Domain                                   |
> |-----------------|------------------------------------------------|------------------------------------------------|
> | DeCLIP                  | **2.81** (5.35 / 4.33 / 1.29 / 0.28)           | **2.97** (5.49 / 4.96 /1.10 / 0.31)           |
> | GroundingDino        | **4.08** (4.49 / 9.33 / 2.29 / 0.21)           | **3.96** (4.26 / 8.97 / 2.41 / 0.19)          |
> | Ins-DetCLIP-FlanT5-base    | **15.3** (24.5 / 15.3 / 11.3 / 10.0)           | **13.7** (24.2 / 16.0 / 8.62 / 5.90)           |
> | Ins-DetCLIP-FlanT5-large | **16.2** (25.6 / 16.4 / 11.7 / 11.0)           | **14.4** (25.4 / 16.2 / 9.65 / 6.50)           |
>
> We observe that directly using the OVD methods for IOD tasks achieve poor performances, since they are not able to interpret natural language instructions.
>
> Therefore, we choose the 2-stage sequential baselines due to the following reason: the Multimodal LLMs have shown superior ability in terms of instruction-following and reasoning, which makes it natural to utilize them to first name all the objects of interest given the instruction, then perform localization with the open-vocabulary detectors.
>
> However, we agree that the two-stage approaches are indeed sub-optimal, which suffer from the following problems:
> - input images of the VIT encoder typically have low resolution, which makes them unable to identify small objects;
> - the MLLM are known to suffer from hallucination, which often mention objects that do not appear in the image;
> - the LLM encounter difficulty in naming all the relevant objects sequentially when the number of objects grows large.
>
> This is where our approach demonstrates advantage, since 1) we adopt DetCLIP to extract fine-grained object level features from high-resolution images, which prevents  the issue with small objects;  2) the LLM directly performs reasoning directly for each object feature in parallel, which reduces the difficulty for generation, achieves better results and boosts the efficiency.
>
> Overall, many thanks for your valuable comments.  We have updated our paper with the experiments and detailed explanations accordingly. We hope our answers have addressed each of your concerns. If you have any further questions, we are happy to address them.

---

> ### Author Response · Authors · 2023-11-21
> **Waiting for further discussion.**
>
> Dear Reviewer Y4Gg,
>
> We are grateful for your valuable time during the reviewing process, and we appreciate all your constructive comments on our work. We are also very glad that you acknowledge that our work solves real-world problems through diverse instructions to be interesting.
>
> Since it has come to the end of the first rebuttal phase, we sincerely hope that our response has addressed your problems and will continue to address further questions if you have any.
>
> Thank you!

---

> ### Author Response · Authors · 2023-11-23
> **Waiting for further discussion.**
>
> Dear Reviewer Y4Gg,
>
>
> We really appreciate your effort in reviewing our paper and your acknowledgement that our idea is interesting again. In the previous section, we have updated our draft and added replies to your comments with our latest experimental results (i.e.,  add clarifications for the design of our IOD tasks, add experiments to compare Ins-DetCLIP with open-vocabulary detectors).
>
> As the end of the discussion period is approaching this evening, we would appreciate it if you could let us know whether our responses have addressed your concerns satisfactorily. If your concerns have not been resolved, could you please let us know about it so that we have the opportunity to respond before the deadline? Or if you are satisfied with our response, we would appreciate it very much if you could consider updating your rating.
>
> We understand you are very busy and we really appreciate your time. We will look forward to your valuable feedback. Thank you!
>
>
> Best wishes,
>
> Paper4489 Authors

---

### Official Review · Reviewer_tem2 · 2023-10-31

**Soundness:** 3 good
**Presentation:** 3 good
**Contribution:** 3 good
**Rating:** 6
**Confidence:** 4

**Summary:**

This paper introduces Instruction-oriented Object Detection (IOD), a novel task aimed at improving human-computer interaction by enabling object detectors to interpret user instructions for identifying specific objects. IOD necessitates the understanding of natural-language instructions and contextual reasoning to provide the name and location of the desired objects, posing new challenges to current object detection systems. To address this, the authors develop a dataset called IOD-Bench, consisting of instruction-guided detections and specialized evaluation metrics, and leverage large-scale language models (LLMs) to generate a diverse set of instructions based on existing public object detection datasets. The proposed model, Ins-DetCLIP, utilizes the knowledge within LLMs to enable instruction-following capabilities in the detector. It employs a visual encoder, DetCLIP, to extract object-level features and aligns them with the input instructions through a cross-modal fusion module integrated into a pre-trained LLM. The experimental results on IOD-Bench demonstrate that Ins-DetCLIP consistently outperforms baseline methods.

**Strengths:**

1. Overall, the task presented in this manuscript is captivating and aligns well with the practical applications of an intelligent detection system. The instructions have been well-crafted, bolstering my view on this matter.

2. The clarity of the writing in this submission is commendable, making the content generally straightforward to grasp.

3. The impressive performance of Ins-DetCLIP underscores the effectiveness of the proposed instruction tuning paradigm. Furthermore, the authors have done a good providing an extensive range of experiments to scrutinize all the design choices, which is highly valuable.

**Weaknesses:**

1. I would recommend that the authors take another pass at proofreading the manuscript to ensure consistent and correct formatting throughout. For instance, there is a conventional practice of inserting a space before references in both the main body text and tables.

2. With respect to the main results showcased in Table 1, while they are compelling and seem to align with the authors' motivations, I note that the comparison methods, such as BLIP-2 and MiniGPT-4, do not incorporate human instructions interns of object detection. I presume that performing instruction tuning on these models could enhance them and provide insights into the generalizability of the proposed tuning method. I am curious about the feasibility of this approach.

3. In terms of the dense captioning tasks, despite the impressive performance demonstrated, the comparison is made with somewhat outdated methods. There are contemporary methods, building on SAM or other models, capable of performing this task as well. A performance comparison with Ins-DetCLIP would be insightful. While Ins-DetCLIP may not outperform these methods, including such results or providing justification would help to elucidate the performance gap, potentially laying the groundwork for future enhancements.

**Questions:**

Please refer to the weaknesses.

---

> ### Author Response · Authors · 2023-11-17
> **Response to Reviewer tem2**
>
> Thank you very much for your insightful suggestions. We are glad that you find our proposed task to be captivating and align well with the practical applications of an intelligent detection system, the instructions are well crafted, and the experiments to be thorough. We wish to address your concerns by giving detailed responses to each of your comments as follows:
>
> **1. Reference formats are wrong.**
>
> Thank you so much for pointing out this careless mistake. We have corrected the citations in the updated version.
>
>
> **2. Instruction tuning on the two-stage baselines.**
>
> Thank you for pointing out this potential confusion. For experiments in Table 1, We indeed perform instruction tuning on the LLM part of the two-stage baselines, which aims to enable the LLM to elicit the target object categories corresponding to the user's instruction. Such categories are then given to the open-vocabulary object detector (OVOD) to perform localization. For completeness, we further compare with the two-stage baselines without instruction tuning in the table below, and update it to Table 8 of the Appendix. The results show that tuning with our dataset also boosts the performances of two-stage baselines, which proves instruction tuning indeed generalizes well to other approaches. However, due to the design choices, such methods are still inferior to end2end instruction tuning of Ins-DetCLIP:
>
> The reasons for the inferior performance of two-stage baselines in IOD tasks are as follows:
> - Input images of the VIT encoder typically have low resolution, which makes them unable to identify small objects. In contrast, the input resolution of our DetCLIP backbone has high resolution, which prevents the problem;
> - The LLMs are known to suffer from hallucination, which often mention objects that do not appear in the image;
> - The LLM encounter difficulty in naming all the relevant objects sequentially when the number of objects grows large.
>
> | MLLM                                   | Detector | Tuned | In domain                                     | Out of Domain                                   |
> |----------------------------------------|----------|-------|-----------------------------------------------|------------------------------------------------|
> | BLIP2-FlanT5-XL | DetCLIP | N     | 3.06 (4.16 / 3.95 / 2.34 / 1.80)           | 3.03 (4.27 / 3.70 / 2.45 / 1.68)               |
> | BLIP2-FlanT5-XL          | DetCLIP  | Y     | 3.95 (5.37 / 4.51 / 3.04 / 2.89)           | 3.71 (5.21 / 4.40 / 2.79 / 2.43)               |
> | MiniGPT4-Vicunna-7B| DetCLIP  | N     | 5.57 (10.5 / 7.20 / 2.62 / 1.94)           | 5.45 (9.63 / 7.31 / 2.69 / 2.18)               |
> | MiniGPT4-Vicunna-7B| DetCLIP  | Y     | 8.29 (12.3 / 10.3 / 7.51 / 3.05)           | 6.35 (10.4 / 7.13 / 5.29 / 2.57)               |
> | Ins-DetCLIP-Opt1.3b                     | DetCLIP  | Y     | **14.9 (22.9 / 14.7 / 11.5 / 10.4)**          | **11.4 (20.4 / 13.6 / 7.42 / 4.10)**           |
> | Ins-DetCLIP-FlanT5-base                 | DetCLIP  | Y     | **15.3 (24.5 / 15.3 / 11.3 / 10.0)**          | **13.7 (24.2 / 16.0 / 8.62 / 5.90)**           |
> | Ins-DetCLIP-FlanT5-large                | DetCLIP  | Y     | **16.2 (25.6 / 16.4 / 11.7 / 11.0)**          | **14.4 (25.4 / 16.2 / 9.65 / 6.50)**           |
>
>
>
> **3. Comparison with more up-to-date methods on dense-captioning tasks.**
>
> Thank you for pointing this out. We further compare with more contemporary methods on dense caption tasks, which include CapDet (CVPR2023) [1] and Grit (Arxiv) [2]. Among them, CapDet is based on the image-language foundation model BLIP. The results are shown in the following table, where our method still achieves slightly better performance. Even though we were not able to discover new dense captioning methods that are based on powerful vision experts such as SAM, it is indeed a viable approach that could lead to strong performance. We will leave this investigation to future research. We also updated this comparison in Table 1 of the main paper.
>
> | Method                    | VG V1.2 mAP (%) | Method                    | VG COCO mAP (%) |
> |---------------------------|-----------------|---------------------------|-----------------|
> | CapDet   | 15.44 | CapDet  | 13.98 |
> | Grit    | 15.50 | Grit |  -    |
> | Ins-DetCLIP-Flanbase      | **15.70**       | Ins-DetCLIP-Flanbase      | **14.35**       |
> | Ins-DetCLIP-Flanlarge     | **16.13**       | Ins-DetCLIP-Flanlarge     | **15.01**       |
>
> Overall, many thanks for your insightful points and suggestions. We have added the experiments and revised our paper accordingly in our latest version.  We hope our answers have addressed your concerns. If you have any further questions, we are happy to address them.
>
> **References:**
>
> [1] CapDet: Unifying Dense Captioning and Open-World Detection Pretraining
>
> [2] GRiT: A Generative Region-to-text Transformer for Object Understanding

---

> > ### Comment · Reviewer_tem2 · 2023-11-21
> > **Reply to the authors' rebuttal**
> >
> > Many thanks for your comprehensive and prompt rebuttal. I think the experiments that you provided are convincing. I strongly recommend adding them to the main paper.
> >
> > With respect to the experiments on adding instruction tuning to BLIP2 and MiniGPT4, why would BLIP2-FlanT5-XL and Ins-DetCLIP-FlanT5 have such a huge performance gap although they adopt almost identical LLMs?

---

> ### Author Response · Authors · 2023-11-21
> **Response to Reviewer tem2**
>
> Thank you very much for your response! Since we need to stick to the original requirement for 9-page-limit, we will carefully rearrange the paper's contents and update the new results to the main paper in the final version.
>
> **Why would BLIP2-FlanT5-XL and Ins-DetCLIP-FlanT5 have such a huge performance gap although they adopt almost identical LLMs?**
>
> Thank you for this interesting question, the reasons are as follows:
>
> - For the 2-stage baselines, the LLM needs to output all the object names that satisfy the user's instruction. This is particularly challenging, especially for weaker LLMs such as FlanT5-XL. We observe that they tend to miss many objects in the generated response. On the other hand, our method processes each object in parallel, and the LLM only needs to predict the category for a single object, which greatly alleviates the difficulty;
>
> - For two stage baselines, the input images to the MLLM (e.g., BLIP2) typically have low resolution, which makes them unable to identify small objects. In contrast, the input resolution of our DetCLIP backbone has high resolution, which prevents the problem.
>
> Thank you again for your valuable time spent on reviewing our paper.

---

### Official Review · Reviewer_rzsX · 2023-11-04

**Soundness:** 3 good
**Presentation:** 3 good
**Contribution:** 4 excellent
**Rating:** 8
**Confidence:** 3

**Summary:**

The manuscript presents a new task called Instruction-oriented Object Detection (IOD), which aims for the model to accept human instruction and generate corresponding detection results. For training the model on such a task, the authors first construct a dataset termed IOD-Bench and the corresponding evaluation metrics. Based on DetCLIP, Ins-DetCLIP is presented for making the open-vocabulary object detector able to follow instructions, where an LLM is attached to the frozen visual encoder of pretrained DetCLIP.

Empirically, Ins-DetCLIP notably outperforms baseline approaches such as BLIP-2, MiniGPT-4 and LLaVA. Additionally, the model shows the capability of performing captioning for the relevant objects.

**Strengths:**

- The manuscript goes beyond image-level captioning/question answering and proposes to use LLM for instructed object detection, which is novel.
- The approach provides new insights into what a LLM could do when it is connected to an open-vocabulary object detector.
- Compared to existing approaches such as BLIP-2, MiniGPT-4 and LLaVA, Ins-DetCLIP demonstrates an outstanding capability of performing instruction guided object detection.
- Increasing the size of LLM could benefit all the tasks, showing good scalability.

**Weaknesses:**

- It is unclear how the object bounding boxes and the features are generated in the first place before object-level cross modal fusion.
- It is not shown to what extent is the approach dependent on the performance/quality of the phase-1 training, which would be an important aspect for understanding the approach that requires two-phase training.
- The inference speed comparison seems unfair, since the model sizes are different. It is difficult to harness whether the approach is slow or fast.
- Citation formats are wrong.

**Questions:**

- It would be interesting to know how much resources and GPU days are required to train such a model.

---

> ### Author Response · Authors · 2023-11-17
> **Response to Reviewer rzsX**
>
> We sincerely thank Reviewer rzsX for your positive feedback and are grateful for the time you spent on our submission. We are also glad you think our idea is novel and provides new insights. Below we would like to give detailed responses to each of your comments.
>
> **1. How the object bounding boxes and the features are generated before object-level cross-modal fusion.**
>
> Thank you for pointing out this potential confusion. To enable generating the object features and bounding boxes, we introduce a foreground-background classification head, similar to the spirit of the Region Proposal Network (RPN) [1]. In our original submission, we located the corresponding details in the appendix due to the space limitation in our initial submission. However, your comment prompted us to realize that we have missed this important detail in the main paper. We have moved this information to Section 4.2 of the updated paper.
>
> **2. The importance of the phase-1 training.**
>
> Thank you for the suggestion! In order to validate the significance of the phase-1 training, we conduct experiment by engaging in the second stage of instruction tuning without pretraining. Note that we continue to employ ImageNet-pretrained Swin-Transformer weights for initializing the vision backbone, alongside a FILIP-pretrained text encoder in this experiment.  Our observations indicate a substantial decline in the model's performance, primarily attributed to its diminished capability in extracting high-quality object features. More specifically, the model's performance experienced a marked reduction, dropping from 15.3 to 8.46 for in-domain instructions, and from 13.7 to 8.00 for out-of-domain instructions. These findings have been incorporated into Table 9 in the Appendix of our revised manuscript.
>
> | Model                    | Pretrain | In domain                                      | Out of Domain                                    |
> |--------------------------|----------|------------------------------------------------|-------------------------------------------------|
> | Ins-DetCLIP-FlanT5-base  | N        | **8.46** (12.6 / 10.5 / 7.16 / 3.53)            | **7.75** (11.3 / 10.3 / 6.10 / 3.31)            |
> | Ins-DetCLIP-FlanT5-base  | Y        | **15.3** (24.5 / 15.3 / 11.3 / 10.0)            | **13.7** (24.2 / 16.0 / 8.62 / 5.90)            |
>
> **3. The inference speed comparison seems unfair.**
>
> Great suggestion! Indeed, the LLMs used in the two-stage baselines (size > 7B ) are larger than the ones used in Ins-DetCLIP (size < 1B). This is because such methods rely heavily on the quality of  LLM's generated object names, switching to a small model hurts the performance greatly. As suggested, we further show the performance and inference speed of a smaller counterpart, where the performance is severely degraded from 4.32 to 3.14 mAP (updated in Table 1 of the main paper), and the inference speed (1.4 fps) is still slower than Ins-DetCLIP-FlanT5-Base (2.2 fps) and Ins-DetCLIP-FlanT5-Large (1.6 fps), which is updated in Table 7 of the Appendix and shown by the table below. This is because the two-stage baselines require the LLM to first sequentially output all the category names of the objects before providing them to the OV detector. On the other hand, Ins-DetCLIP handles the generation for each target object in parallel.
>
> | Method                 | Inference Speed (FPS) |
> |------------------------|----------------------|
> | Blip2-FlanBase         | 1.4                  |
> | Ins-DetCLIP-FlanBase   | **2.2**              |
> | Ins-DetCLIP-FlanLarge  | **1.6**              |
>
> **4. Citation formats are wrong.**
>
> Thank you so much for pointing out this careless mistake. We have corrected the citations in the updated version.
>
> **5. Resources and GPU days required to train the model.**
>
> For the first stage, the training of DetCLIP-T with Swin-T backbone takes 63 hours on 32 V100 GPUs. The resulting model from the pretraining stage can already serve as a powerful open-vocabulary object detector. In the second stage, it takes only around 24 hours on 16 V100 GPUs for instruction tuning. We have added this information in the Appendix of our updated paper.
>
> Thank you very much for the constructive comments, which really help us further improve our work. We hope our answers have addressed your concerns. If you have any further questions, we are happy to address them
>
> **References:**
>
> [1] Faster R-CNN: Towards Real-Time Object Detection with Region Proposal Networks

---

### Author Response · Authors · 2023-11-17
**Message To All Reviewers**

We thank all the reviewers for their valuable time, constructive suggestions, and insightful comments. We appreciate that the reviewers agree that **our proposed IOD task is interesting and practical** (Reviewer rzsX, tem2, Y4Gg, t19r), **the idea of using LLM  for instruction-oriented detection is novel** (Reviewer rzsX, tem2), **experiment results are thorough and insightful** (Reviewer rzsX, tem2), and **the method is simple and easy to understand (Reviewer t19r)**.

In this rebuttal, we aim to address the concerns and confusions raised by the reviewers.

We summarize the updates we made on the paper during the rebuttal period:

**(1) In response to Reviewer rzsX**:
- In Section 4.2 of the main paper, we add elaboration for the generation of object bounding boxes;
- In Table 9 of the Appendix, we add an experiment by skipping the pretraining stage, demonstrating the importance of the first stage pretraining.
- In Table 7 of the Appendix, we further compare the inference speed with the baseline model having similar size as Ins-DetCLIP.
- In Section B of the Appendix, we elaborate on the training cost of our model.
- We fix the citation format.

**(2) In response to Reviewer tem2:**
- In the Table 8 of the Appendix, we further compare with the two-stage baselines w/o instruction tuning.
- In Table 2 of the main paper, we compare with more recent dense captioning tasks.
- We fix the citation format.

**(3) In response to Reviewer Y4Gg:**
- In Section C of the Appendix, we add clarifications for the design of our IOD tasks.
- In Table 11 at the Appendix, we compare Ins-DetCLIP with open-vocabulary detectors on our IOD task.

**(4) In response to Reviewer t19r:**
- In Section C of the Appendix, we elaborate the definition of task 4.
- In the related work section, we add citations to InstructBLIP.
- In Table 10 in the Appendix, we make comparisons with Shikra and Kosmos-2.
- We discussed why using off-the-shelf detectors and LLM cannot achieve good results.

At the current rebuttal phase,  we update most of the new contents in the Appendix to prevent surpassing the 9-page limit. We will re-organize the paper and incorporate those contents to the main section of the paper.

---

### Author Response · Authors · 2023-11-20
**Waiting for further discussion.**

Dear all reviewers,
We sincerely thank all the reviewers for your constructive comments and insightful suggestions, which help us make our work more complete and further improve the quality of the manuscript. We are also glad that the reviewers acknowledge that our work 1) **proposes an IOD task that is interesting and practical**, 2) **the idea is novel and easy to understand**, and 3) **the experiments are thorough and insightful**. According to the advice, we have made the following changes in the current version of our paper (highlighted in red):
- **(1) In response to Reviewer rzsX**:
    - In Section 4.2 of the main paper, we add elaboration for the generation of object bounding boxes;
    - In Table 9 of the Appendix, we add an experiment by skipping the pretraining stage, demonstrating the importance of the first stage pretraining.
    - In Table 7 of the Appendix, we further compare the inference speed with the baseline model having similar size as Ins-DetCLIP.
    - In Section B of the Appendix, we elaborate on the training cost of our model.
    - We fix the citation format.

- **(2) In response to Reviewer tem2**:
    - In the Table 8 of the Appendix, we further compare with the two-stage baselines w/o instruction tuning.
    - In Table 2 of the main paper, we compare with more recent dense captioning tasks.
    - We fix the citation format.

- **(3) In response to Reviewer Y4Gg**:
    - In Section C of the Appendix, we add clarifications for the design of our IOD tasks.
    - In Table 11 at the Appendix, we compare Ins-DetCLIP with open-vocabulary detectors on our IOD task.

- **(4) In response to Reviewer t19r**:
    - In Section C of the Appendix, we elaborate the definition of task 4.
    - In the related work section, we add citations to InstructBLIP.
    - In Table 10 in the Appendix, we make comparisons with Shikra and Kosmos-2.
    - We discussed why using off-the-shelf detectors and LLM cannot achieve good results.


**Thank you again for the effort in reviewing our paper. As the end of the discussion period is approaching, we sincerely and humbly ask for your response. We would appreciate it if you could let us know whether our responses have addressed your concerns satisfactorily and whether you have any follow-up questions.**

Best regards,

Paper 4489 Authors

---

### Author Response · Authors · 2023-11-22
**Waiting for further discussion.**

Dear Reviewers,

Thank you for your valuable feedback and insightful suggestions, which have significantly contributed to improving our manuscript. As the discussion period nears its end, we welcome any further comments or questions you may have. Your continued engagement is highly appreciated.

Best regards,

Paper 4489 Authors

---

### Public Comment · ~Abhishek_Aich1 · 2024-03-15
**Code release?**

Hi authors,

Congrats on your paper acceptance and the great work. Do you plan to release the training/testing code?

---

### Meta-Review · Area_Chair_mj3M · 2023-12-05

**Metareview:**

This paper receives mixed reviews initially. Some reviewers are positive while others are negative. The raised issues include unclear technical details, insufficient experimental validations, the unclear relationship upon existing models. During the rebuttal phase, the authors provide sufficient justifications and experiments to address these issues. Overall, the AC monitors the whole progress and feels the major issues are solved. The authors shall revise the paper according to these comments.

**Justification For Why Not Higher Score:**

The designed pipeline is built up based on existing frozen models.

**Justification For Why Not Lower Score:**

The performance is promising and the rationale is new.

---

### Decision · Program_Chairs · 2024-01-16

Accept (poster)